# Autoxidation of 4-Hydrazinylquinolin-2(1*H*)-one; Synthesis of Pyridazino[4,3-*c*:5,6-*c*′]diquinoline-6,7(5*H*,8*H*)-diones

**DOI:** 10.3390/molecules27072125

**Published:** 2022-03-25

**Authors:** Sara M. Mostafa, Ashraf A. Aly, Alaa A. Hassan, Esraa M. Osman, Stefan Bräse, Martin Nieger, Mahmoud A. A. Ibrahim, Asmaa H. Mohamed

**Affiliations:** 1Chemistry Department, Faculty of Science, Minia University, El Minia 61519, Egypt; sara.ahmed@mu.edu.eg (S.M.M.); alaahassan2001@mu.edu.eg (A.A.H.); esraamah33@gmail.com (E.M.O.); m.ibrahim@compchem.net (M.A.A.I.); asmaa.hamouda@mu.edu.eg (A.H.M.); 2Institute of Organic Chemistry, Karlsruhe Institute of Technology, Eggenstein-Leopoldshafen, 76131 Karlsruhe, Germany; 3Institute of Biological and Chemical Systems (IBCS-FMS), Karlsruhe Institute of Technology, Eggenstein-Leopoldshafen, 76131 Karlsruhe, Germany; 4Department of Chemistry, University of Helsinki, P.O. Box 55 (A. I. Virtasen aukio I), 00014 Helsinki, Finland; martin.nieger@helsinki.fi

**Keywords:** 4-hydrazinylquinolin-2(1*H*)-one, pyridazino[4,3-*c*:5,6-*c*′]diquinoline-6,7(5*H*,8*H*)-dione, autoxidation reaction, X-ray, DFT

## Abstract

An efficient synthesis of a series of pyridazino[4,3-*c*:5,6-*c*′]diquinolines was achieved via the autoxidation of 4-hydrazinylquinolin-2(1*H*)-ones. IR, NMR (^1^H and ^13^C), mass spectral data, and elemental analysis were used to fit and elucidate the structures of the newly synthesized compounds. X-ray structure analysis and theoretical calculations unequivocally proved the formation of the structure. The possible mechanism for the reaction is also discussed.

## 1. Introduction

Compounds with quinolin-2(1*H*)-one (carbostyril) skeletons are present in a large number of biologically active compounds [1,2,3,4,5,6,7,8,9,10,11,12]. During the twentieth century [13], various research groups dealt with the chemistry and biological applications of quinolines [14].

Pyridazine and its polycyclic structures have still played an interesting role in organic chemistry because of their remarkable properties in forming supramolecular assembly [15,16,17,18]. Pyridazine molecules are important heterocycle scaffolds that reveal diverse biological activities in medicine [19,20,21,22,23,24,25] and agriculture [26]. For example, Pyridazomycin is an antifungal and antibiotic compound, the first pyridazine derivative isolated from a natural source [27]. In contrast, Pyridaben is widely used as an acaricide with a long residual action, whereas Chloridazone has a long history of use as an herbicide [28]. Minaprine is a psychotropic drug that has effectively treated various depressive states [29] (Figure 1).

Functionalized pyridazines have high electron-deficient properties, encouraging their utilization as electrochromic materials and metal–organic frameworks [30,31].

Hydrazines have shown basic and reducing characteristics that enable their utility in many industrial and medical applications. Accordingly, hydrazines have served as rocket fuel, antioxidants, oxygen scavengers, and as intermediates for the production of explosives, propellants, and pesticides [32].

It has been demonstrated that oxygen makes hydrazine solutions unstable, especially under alkaline or neutral conditions. However, hydrazines are stable under strongly acidic conditions or without oxygen [33].

Wagnerova et al. [34] have reported that cobalt-tetrasulphophthalocyanines enhance the autoxidation process of hydrazines [34]. On the other hand, Ichiro Okura et al. [35] reported that the autoxidation of hydrazine occurred with manganese (III)-hematoporphyrin at room temperature, and that has an advantage because of its reactivity for the formation of oxygen coordinated Mn(III)-Hm peroxide adduct [36]. Andrew P. Hong et al. [37] reported that satisfactory autoxidation of hydrazines occurred using cobalt (II) 4,4′,4″,4‴-tetrasulfophthalocyanine (CoIITSP).

Previously, it was reported [38] that 1-ethyl-4-hydroxyquinolin-2(1*H*)-one (**1**) reacted with hydrazine hydrate, in 1,2-dichlorobenzene (*o*-DCB), to give a mixture of two compounds. These compounds were separated using fractional recrystallization to give quinolinylhydrazine **2a** in 19% yield and diquinopyridazine **3a** in 41% yield (Figure 1).

A convenient microwave-assisted, one-pot, four-component synthetic approach was developed as a route to functionalized benzo[a]pyridazino[3,4-c]phenazine derivatives starting from 2-hydroxy-1,4-naphthoquinone, aromatic aldehydes, methyl hydrazine, and *o*-phenylenediamine. Compounds of a similar pentacyclic structure such as bisanthranilate showed an intramolecular electrophilic cyclization and afforded an angular *cis*-quinacridone compound, which condensed with hydrazine to give a phthalazine derivative [39]. The biological profiles of some of the compounds mentioned above exhibited good cytotoxic activities against KB, HepG2, Lu1, and MCF7 human cancer cell lines. In addition, a compound of the derivatives exhibited promising antimicrobial activities toward *Staphylococcus aureus* and *Bacillus subtilis* bacterial strains with IC_50_ < 6 μM [40].

Recently, it has been reported that hydrazines can be used as catalysts for removing oxygen in the closing carbonyl–olefin metathesis process [41]. We have also found that prolongated reflux during the formylation process of 2-quinolones *via* a DMF/Et_3_N mixture caused dimerization to occur, and unexpected 3,3′-methylenebis(4-hydroxyquinolin-2(1*H*)-ones) were obtained [42].

The above-mentioned findings encouraged us to generalize the method of preparation for heteroannulated pentacyclic compounds with the structure of this interesting molecule.

## 2. Results and Discussion

The strategy started with the preparation of derivatives of compounds **1**, **2**, **4**, and **5** according to reported methods, and their structures were confirmed by matching their spectral data with those reported [43,44,45]. The key intermediates, hydrazine quinolones **2a**–**g**, were prepared by refluxing compounds **4a**–**g** with hydrazine hydrate (Figure 2) [46]. During the heating of 4-hydrazinylquinolin-2(1*H*)-ones **2a**–**g** in pyridine, we observed the abnormal formation, in good yields, of pyridazino[4,3-*c*:5,6-*c*′]diquinoline-6,7- (5*H*,8*H*)-diones **3a**–**g**. As had been suggested, compounds **2a**–**g** underwent an autoxidation reaction.

The structures of the products **3a**–**g** were proved from their elemental analyses and IR, ^1^H NMR, and ^13^C NMR spectra. For example, the mass spectrum and elemental analysis of **3a** established its molecular formula as C_22_H_18_N_4_O_2_. The ^1^H NMR spectrum of **3a** exhibited the ethyl protons as a triplet at *δ**_H_* = 1.22 (*J* = 7.6 Hz) for CH_3_ and a quartet at *δ**_H_* = 4.39 ppm (*J* = 7.6 Hz) for CH_2_. The eight aromatic protons appeared as three multiplets at *δ**_H_* = 7.36–7.40 for 2H, 7.68–7.78 for 4H, and 8.06–8.08 ppm for 2H. The reported spectroscopic data for the ^13^C NMR spectrum of compound **3a** showed the carbonyl-quinolone, 2NCH_2,_ and CH_3_ carbon signals at *δ_C_* = 165.72, 39.11, and 14.11 ppm, respectively. Similar spectroscopic results of compound **3a** were also reported [38]. The structure of **3a** was unambiguously determined by a single crystal structure (Figure 2).

We carried out the reaction in different conditions using compound **1a** as an example with the optimized reaction conditions. In EtONa/EtOH (Method **B**, Table 1), it was found that the yield of **3a** was decreased (74%). Refluxing of **1a** in toluene/Et_3_N (Method **C**, Table 1) did not increase the yield (60%), and the time taken to obtain **3a** was increased (2d). Furthermore, adding a few drops of Et_3_N to DMF (Method **D**, Table 1) improved the yield of **3a** compared with methods **B** and **C**, but it was still lower compared with our method **A**. Using Na/toluene, the oxidation of **1a** occurred satisfactorily; however, it was lower compared with method **A**. In our trial of an acidic medium using HCl/EtOH mixture, the reaction failed. Thus, the best condition to obtain high yields and a short reaction time of **3a**–**g** is reflux in dry pyridine (Method **A**, Table 1).

The formation of pyridazino[4,3-c:5,6-c′]diquinoline-6,7(5*H*,8*H*)-diones **3a**–**g** can be rationalized as depicted in Figure 3. It is clear from the suggested mechanism (Figure 3) that it constitutes several steps of nucleophilic substitution, dimerization, autoxidation, and electrocyclic reactions in a one-pot process leading to the pentacyclic final products **3a**–**g**. The mechanism starts with a proton shift of compound **2** to its isomer **2′** (Figure 3). Then, the starting molecule of **2** reacts with its isomer **2′** to give **6** (Figure 3). The elimination of a hydrazine molecule in **6** would give the dimerized hydrazone **7,** which undergoes another proton shift to give the intermediate **7′**. Because the reaction did not proceed under an inert argon atmosphere (i.e., under argon atmosphere, the starting quinolinyl-hydrazines **2a**–**g** were recovered), we proposed that the intermediate **7′** undergo an aerial oxidation NH-NH group to give the intermediate **8**. After that, the intermediate **8** would undergo internal electrocyclization to give **9**. Finally, another mode of aerial oxidation of **9** would produce compound **3** (Figure 3).

The preceding literature supports the mechanism [47] describing aryl hydrazine chlorides’ aerial oxidation into diazines. Accordingly, it supports the steps of transformations of **7′** into **8** and **9** into **3**. Moreover, aerial catalytic oxidation in pyridine transformed hydrazones into diazo compounds [48].

Firstly, the stability of compound **3a** (Figure 3), as an example, was described. Therefore, the quantum mechanical calculations were performed for compound **3a**. The investigated compound was first optimized using the DFT method (see the Methods section for details). The optimized structure was then subjected to vibrational frequency and single-point energy calculations. The quantum theory of atoms in molecules (QTAIM) was invoked to achieve an in-depth insight into the topological features of compound **3a** [49]. In the context of QTAIM, the (3,–1) bond critical points (BCPs) and bond paths (BPs) were generated, and the electron density was computed. Moreover, noncovalent interaction (NCI) index analysis was executed to pictorially elucidate the origin and nature of intramolecular interactions within compound **3a** [50]. According to the results, no imaginary frequencies were observed for the investigated structure of compound **3a**, confirming that this conformer is a true minimum. Based on the QTAIM results presented in Figure 4a, the occurrence of intramolecular bonds within the inspected compound was revealed by the existence of BPs and BCPs. Chalcogen∙∙∙chalcogen intramolecular interaction was also noticed in compound **3a** via the BP and BCP between the two oxygen atoms (O∙∙∙O). The BCP at the BP O∙∙∙O within compound **3a** exhibited electron density with a value of 0.0144 au.

The stability of compound **3a** might also be interpreted as a consequence of the aromatic planarity, which could be detected from Figure 4a *via* dihedral angles (*Φ*) with a value of 1.83°. Notably, the difference between the dihedral angle of the optimized geometry of compound **3**a and the X-ray data was nearly 0.36°.

As shown in Figure 4b, the NCI results (green isosurfaces) occurred at the interatomic space between the interacting atoms, asserting the occurrence of the intramolecular interactions towards the investigated compound. Large, green, round domains within the intramolecular forces N_13_∙∙∙HC_12_ and N_14_∙∙∙HC_1_ of compound **3****a** were crucially denoted, reflecting the favorable contribution of such intramolecular forces to the further stability of compound **3a**.

## 3. Conclusions

The unprecedented dimerization and oxidation cascade of 4-hydrazinylquinolin-2(1*H*)-ones delivered pyridazino[4,3-*c*:5,6-*c*′]diquinoline-6,7(5*H*,8*H*)-diones in good yields. The synthesis of the obtained pyridazino-diquinolones was achieved in different conditions. Heating 4-hydrazinylquinolin-2(1*H*)-ones in pyridine was the best condition in which to obtain the corresponding products. Quantum mechanical calculations were also performed using the DFT method to prove the stability of the formed products. The method above can be used as a general method for the preparation of various classes of pentacyclic heterocycles from compounds with structural features similar to 4-hydroxy-2-quinolones. Importantly, the biological activity of the obtained products could assist in the development of new drugs.

## 4. Experimental Section

### 4.1. Chemistry

The IR spectra were recorded using the ATR technique (ATR = Attenuated Total Reflection) with an FT device (FT-IR Bruker IFS 88), Institute of Organic Chemistry, Karlsruhe University, Karlsruhe, Germany. The NMR spectra were measured in DMSO-*d*_6_ on a Bruker AV-400 spectrometer, 400 MHz for ^1^H, and 100 MHz for ^13^C; the chemical shifts are expressed in *δ* (ppm), versus internal tetramethylsilane (TMS) = 0 for ^1^H and ^13^C, and external liquid ammonia = 0. The description of signals includes: s = singlet, d = doublet, t = triplet, q = quartet, m = multiplet, dd = doublet of doublet, and m = multiplet. Mass spectra were recorded on a FAB (fast atom bombardment) Thermo Finnigan Mat 95 (70 eV). Elemental analyses were carried out at the Microanalytical Center, Cairo University, Egypt. TLC was performed on analytical Merck 9385 silica aluminum sheets (Kieselgel 60) with Pf254 indicator; TLCs were viewed at λmax = 254 nm.

#### 4.1.1. Starting Materials

4-Chloro-quinolin-2(1*H*)-ones **4a**–**g** were prepared according to the literature [43,44,45], whereas 4-hydrazinylquinolin-2(1*H*)-ones **2a**–**g** were synthesized according to the literature [46].

#### 4.1.2. General Procedure

4-Hydrazinylquinolin-2(1*H*)-ones **2a**–**g** (1 mmol) were heated in pyridine for 6–12 h until the reactants had disappeared, as mentioned in Figure 1. The reaction was monitored using TLC (using toluene: EtOAc; 10:1). Then, the mixture was cooled and poured into iced water; the formed precipitate was filtered, washed with water, and recrystallized from the stated solvents to give pure crystals **3a**–**g**.

##### 5,8-Diethylpyridazino[4,3-*c*:5,6-*c*’]diquinoline-6,7(5*H*,8*H*)-dione (**3a**)

Orange crystals (DMF/H_2_O), yield: 0.31 g (86%); mp 330–332 °C. IR (KBr): υ = 3013 (Ar-CH), 2927 (Ali-CH), 1653 (C=O), 1603, 1561 (Ar-C=C) cm^−1^; NMR as reported in ref [38].

##### 5,8-Dimethylpyridazino[4,3-*c*:5,6-*c*’]diquinoline-6,7(5*H*,8*H*)-dione (**3b**)

Orange crystals (DMF/EtOH), yield: 0.30 g (88%); mp 320–322 °C. IR (KBr) υ = 3023 (Ar-CH), 2929 (Ali-CH), 1652 (C=O), 1604, 1586 (Ar-C=C) cm^−1^; ^1^H NMR (400 MHz, DMSO-*d*_6_) *δ**_H_* = 3.72 (s, 6H, CH_3_), 7.38–7.40 (m, 2H, Ar-H), 7.71–7.76 (m, 4H, Ar-H), 8.03–8.06 (m, 2H, Ar-H); ^13^C NMR (100 MHz, DMSO-*d*_6_): *δ**_C_* = 39.91 (2 CH_3_), 115.87, 115.78, 119.12, 122.14, 122.25, 131.66, 136.82, 156.61, 165.81 ppm (C-6 and C-7); MS (Fab, 70 eV, %): *m*/*z* = 342 (M^+^, 82), 289 (6), 197 (13), 171 (23), 169 (36), 157 (100), 134 (48), 105 (15). *Anal. Calcd. for* C_20_H_14_N_4_O_2_ (342.35): C, 70.17; H, 4.12; N, 16.37. Found: C, 70.09; H, 4.23; N, 16.44.

##### Pyridazino[4,3-*c*:5,6-*c*’]diquinoline-6,7(5*H*,8*H*)-dione (**3c**)

Orange crystals (DMF/H_2_O), yield: 0.257 g (82%); mp 348–350 °C. IR (KBr) υ =3174 (NH), 3044 (Ar-CH), 1640 (C=O), 1602, 1585 (Ar-C=C) cm^−1^; ^1^H NMR (400 MHz, DMSO-*d*_6_) *δ**_H_* = 7.05–7.09 (m, 2H, Ar-H), 7.18–7.23 (m, 2H, Ar-H), 7.33–7.38 (m, 2H, Ar-H), 7.68 (dd, 2H, *J* = 7.8, 1.2 Hz, Ar-H), 11.99 (s, 2H, NH); ^13^C NMR (100 MHz, DMSO-*d*_6_): *δ**_C_* = 115.85, 115.96, 119.12, 122.52, 122.78, 130.87, 136.80, 157.77, 165.94 ppm (C-6 and C-7); MS (Fab, 70 eV, %): *m*/*z* = 314 (M^+^, 15), 287 (45), 243 (6), 228 (70), 171 (12), 157 (84). *Anal. Calcd. for* C_18_H_10_N_4_O_2_ (314.30): C, 68.79; H, 3.21; N, 17.83. Found: C, 68.68; H, 3.35; N, 17.77.

##### 2,11-Dimethylpyridazino[4,3-*c*:5,6-*c*’]diquinoline-6,7(*5H*,8*H*)-dione (**3d**)

Orange crystals (DMF/H_2_O), yield: 0.287 g (84%); mp 352–354 °C. IR (KBr) υ = 3178 (NH), 3052 (Ar-CH), 1643 (C=O), 1606, 1581 (Ar-C=C) cm^−1^; ^1^H NMR (400 MHz, DMSO-*d*_6_) *δ**_H_* = 2.37 (s, 6H, CH_3_), 7.20–7.29 (m, 2H, Ar-H), 7.30–7.40 (m, 2H, Ar-H), 7.70 (d, 2H, *J* = 1.7 Hz, Ar-H), 12.21 (s, 2H, NH); ^13^C NMR (100 MHz, DMSO-*d*_6_): *δ**_C_* = 20.59 (2 CH_3_), 115.77, 115.88, 119.12, 122.14, 131.66, 132.15, 134.82, 157.62, 165.71 ppm (C-6 and C-7); MS (Fab, 70 eV, %): *m*/*z* = 342 (M^+^, 25), 327 (20), 312 (5), 271 (38), 171 (100), 157 (30). *Anal. Calcd. for* C_20_H_14_N_4_O_2_ (342.35): C, 70.17; H, 4.12; N, 16.37. Found: C, 70.25; H, 4.03; N, 16.25.

##### 2,11-Dichloropyridazino[4,3-*c*:5,6-*c*’]diquinoline-6,7(5*H*,8*H*)-dione (**3e**)

Orange crystals (DMF/H_2_O), yield: 0.325 g (85%); mp 358–360 °C. IR (KBr) υ = 3177 (NH), 3023 (Ar-CH), 1640 (C=O), 1605, 1560 (Ar-C=C) cm^−1^; ^1^H NMR (400 MHz, DMSO-*d*_6_) *δ_H_* = 7.34–7.35 (m, 2H, Ar-H), 7.36–7.37 (m, 2H, Ar-H), 7.68–7.70 (m, 2H, Ar-H), 12.19 (s, 2H, NH); ^13^C NMR (100 MHz, DMSO-*d*_6_): *δ_C_* = 115.78, 115.92, 119.12, 122.10, 128.78, 132.87, 136.80, 157.52, 165.12 ppm (C-6 and C-7); MS (Fab, 70 eV, %): *m*/*z* = 383 (M^+^, 100), 348 (9), 313 (49), 305 (6), 227 (37), 191 (14). *Anal. Calcd. for* C_18_H_8_C_l2_N4O_2_ (383.19): C, 56.42; H, 2.10; N, 14.62. Found: C, 56.51; H, 2.17; N, 14.53.

##### 3,10-Dichloropyridazino[4,3-*c*:5,6-*c’*]diquinoline-6,7(5*H*,8*H*)-dione (**3f**)

Orange crystals (DMF/EtOH), yield: 0.341 g (89%); mp 358–360 °C. IR (KBr) υ = 3180 (NH), 3041 (Ar-CH), 1645 (C=O), 1610, 1567 (Ar-C=C) cm^−1^; ^1^H NMR (400 MHz, DMSO-*d*_6_) *δ_H_* = 7.20–7.35 (m, 2H, Ar-H), 7.37–7.38 (m, 2H, Ar-H), 7.70 (d, 2H, *J* = 7.8 Hz, Ar-H), 12.20 (s, 2H, NH); ^13^C NMR (100 MHz, DMSO-*d*_6_): *δ_C_* = 115.77, 115.88, 119.12, 122.14, 131.12, 132.15, 136.82, 157.62, 164.71 ppm (C-6 and C-7); MS (Fab, 70 eV, %): *m*/*z* = 383 (M^+^, 70), 313 (31) 227 (14), 191 (100), 153 (89), 137 (31), 110 (18). *Anal. Calcd. for* C_18_H_8_C_l2_N_4_O_2_ (383.19): C, 56.42; H, 2.10; N, 14.62. Found: C, 56.30; H, 2.19; N, 14.55.

##### 2,11-Dimethoxypyridazino[4,3-*c*:5,6-*c*’]diquinoline-6,7(5*H*,8*H*)-dione (**3g**)

Orange crystals (DMF/EtOH), yield: 0.325 g (87%); mp 354–356 °C. IR (KBr) υ = 3186 (NH), 3039 (Ar-CH), 1648 (C=O), 1607, 1577 (Ar-C=C) cm^−1^; ^1^H NMR (400 MHz, DMSO-*d*_6_) *δ_H_* = 3.82 (s, 6H, OCH_3_), 7.20–7.23 (m, 2H, Ar-H), 7.32–7.33 (m, 2H, Ar-H), 7.34–7.37 (m, 2H, Ar-H), 12.13 (s, 2H, NH); ^13^C NMR (100 MHz, DMSO-*d*_6_): *δ_C_* = 55.37 (2 OCH_3_), 116.53, 117.54, 120.53, 122.52, 131.30, 132.80, 134.47, 157.77, 165.73 ppm (C-6 and C-7); MS (Fab, 70 eV, %): *m*/*z* = 374 (M^+^, 100), 313 (36), 284 (55), 268 (46), 187 (50), 106 (18). *Anal. Calcd. for* C_20_H_14_N_4_O_4_ (374.35): C, 64.17; H, 3.77; N, 14.97. Found: C, 64.23; H, 3.60; N, 15.09.

#### 4.1.3. Crystal Structure Determination

Single crystals were obtained via recrystallization from DMF/Water. The single-crystal X-ray diffraction study of **3a** was carried out on a Bruker D8 VENTURE diffractometer with a PhotonII CPAD detector at 298 K using Cu-Kα radiation (λ = 1.54178 Å). Dual space methods (SHELXT) [51] were used for structure solution, and refinement was carried out using SHELXL [51] (full-matrix least-squares on F^2^). Hydrogen atoms were localized using difference electron density determination and refined using a riding model (H(N) free). A semi-empirical absorption correction was applied. The absolute structure was determined via the refinement of Parsons’ x-parameters [52].

**3a**: Orange crystals: C_22_H_18_N_4_O_2_, M_r_ = 370.40 g mol^−1^, size 0.20 × 0.12 × 0.04 mm, Orthorhombic, Pca2_1_ (no.29), *a* = 21.1937 (4) Å, *b* = 9.2208 (2) Å, *c* = 17.9646 (3) Å, *V* = 3510.69 (12) Å^3^, *Z* = 8, D_calcd_ = 1.402 Mg m^−3^, F(000) = 1552, μ (Cu-Kα = 0.75 mm^−1^, *T* = 298 K, 36,156 measured reflection (2θ_max_ = 144.40), 6865 independent (R_int_ = 0.057), 506 parameters, 1 restraint, R*_1_* (for 6684 *I* > 2σ(1)) = 0.042, *wR*^2^ (for all data) = 0.115, *S* = 1.02, largest diff. peak and hole = 0.34 eÅ^−3^/−0.22 eÅ^−3^, x = −0.05(11).

CCDC 2,128,843 (**3a**) contains the Appendix A for this paper (see the Appendix A). These data can be obtained free of charge from The Cambridge Crystallographic Data Centre via www.ccdc.cam.ac.uk/data_request/cif (accessed on 10 February 2022).

#### 4.1.4. Theoretical Calculations

Geometrical optimization and vibrational frequency calculations were carried out at the B3LYP/6-31G* level of theory [53,54] for the compounds under consideration. Upon the optimized structures, the energetic features were then evaluated at MP2/6-311 + G** [55]//B3LYP/6-31G* levels of theory. QTAIM and NCI calculations were performed with the help of Multiwfn 3.7 package [56] and were plotted using Visual Molecular Dynamics (VMD) software [57]. All quantum mechanical calculations were carried out at the B3LYP/6-31G* level of theory using Gaussian 09 software (Gaussian, Inc.: Wallingford, CT, USA) [58].

## Data Availability

All pertinent data have been supplied in the Supporting Information that accompanies this article.

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
