# Peer review of "Autoxidation of 4-Hydrazinylquinolin-2(1H)-one; Synthesis of Pyridazino[4,3-c:5,6-c′]diquinoline-6,7(5H,8H)-diones"

_molecules, 2022, doi:10.3390/molecules27072125_

Round 1

Reviewer 1 Report

Some comments have been taken into account and corrected.

The authors did not rewrite the conclusion in such a way that it reflects the main content of the article. The references still need to be refreshed. Only 12 references from 55 date from 2015 and later. In my opinion, this is a very poor indicator reflecting the lack of relevance of the material presented for the scientific community.

Author Response

Answer: Thanks: The conclusion was changed and I added new updated new 15 refs (Red highlights). So that the paper was improved towards refs, conclusion. In addition, English style was also improved.  The mechanism was changed taking into consideration the mechanism of that reported in J Heterocycl. Chem. 2019, 56, 646-650. In experimental section, the J coupling the protons were clarified. All alternations were made in red highlights.

Reviewer 2 Report

This manuscript describes the efficient synthesis of  pyridazino-[4,3-c:5,6-c']diquinolines via autoxidation of 4-hydrazinylquinolin-2(1H)-ones. This work is a continuation of the research of the authors published in J. Heterocycl. Chem. 2019, 56, 646-650. The conditions for the selective preparation of the polycyclic pyridazino[4,3-c:5,6-c']diquinolines were found, and  6 new heterocyclic compounds of this series were prepared. 

The structure of compounds was confirmed with the use of IR and NMR (1H and 13C) spectra, mass spectral data, and elemental analysis.  X-Ray structure analysis was reported for compound 3a.

This work can be recommended for the publication in Molecules after major revision.

1) The mechanism proposed by the authors is controversial. It needs to be further confirmed. Is it possible to isolate or detect quinoline 8? In my opinion, the mechanism proposed in Heterocycl. Chem. 2019, 56, 646-650 is more plausible.

2) What are the possible practical applications of the compounds obtained in this work? In the introduction, it is desirable to write about polycyclic systems bearing pyridazinone moieties and their practical application.

3) Is it possible to carry out a one-pot transformation 4 to 3? This is important for improvement of scientific significance of this work.

4) The authors compare the stability of two isomers 3a and 3a'. But these two compounds can be obtained by completely different mechanisms. It is not appropriate to consider thermodynamic control here. Using the expressions as anti-form and syn-form in this case is not very appropriate because, usually, syn- and anti- are used for diastereoisomers.

Were the authors able to isolate compound 3a'?

5) It is important to expand the scope of obtained structures. For example, can monocyclic 4-hydroxy-2-pyridones be used in this reaction?

6) What methods were used to assign signals in 13C NMR spectroscopy? This should be given in the experimental part.

The authors designate all aromatic protons in the molecules as multiplets. The figures in SI show the presence of doublets and triplets. It is necessary to give a detailed description of the multiplets, indicating the J constants.

Author Response

Answer: The introduction was modified with their references as 15 new updated refs were added (replacing the old refs). The coupling constants of substituted aromatic were written for compounds 3c, 3d and 3f.

 This work can be recommended for the publication in Molecules after major revision.

1) The mechanism proposed by the authors is controversial. It needs to be further confirmed. Is it possible to isolate or detect quinoline 8? In my opinion, the mechanism proposed in J. Heterocycl. Chem. 2019, 56, 646-650 is more plausible.

Answer: The mechanism was totally changed taking into consideration what was reported in J. Heterocycl. Chem. 2019, 56, 646-650

2) What are the possible practical applications of the compounds obtained in this work? In the introduction, it is desirable to write about polycyclic systems bearing pyridazinone moieties and their practical application.

Answer: A paragraph of them was added with its new refs.

3) Is it possible to carry out a one-pot transformation 4 to 3? This is important for improvement of scientific significance of this work.

Answer: The mechanism was changed (i.e. alternations asked by the other reviewers, therefore there is no need for that suggestion (red highlights).

4) The authors compare the stability of two isomers 3a and 3a'. But these two compounds can be obtained by completely different mechanisms. It is not appropriate to consider thermodynamic control here. Using the expressions as anti-form and syn-form in this case is not very appropriate because, usually, syn- and anti- are used for diastereoisomers.

Answer: Since formation of compound 3’a would be in a different mechanism and according to the demands of the other reviewers to omit the possibility of its formation, we only concentrated on the stability of compound 3a and neglect the other possibility of 3’a.

Were the authors able to isolate compound 3a'?

Answer: The suggestion describes the formation of 3’a was removed.

5) It is important to expand the scope of obtained structures. For example, can monocyclic 4-hydroxy-2-pyridones be used in this reaction?

Answer: I thank the reviewer to say that he wants to generalize the idea, however our aim was concerned with 4-hydroxy-2-quinolone, so that would be a target by us in another work

6) What methods were used to assign signals in 13C NMR spectroscopy? This should be given in the experimental part.

Answer: From two sources; one include what was reported by the previous published paper and other by the calculations of some programs such as Chemdraw.

The authors designate all aromatic protons in the molecules as multiplets. The figures in SI show the presence of doublets and triplets. It is necessary to give a detailed description of the multiplets, indicating the J constants.

Answer: The reviewer has a right, especially for substituted aromatic rings. So I added them in compounds 3c, 3d and 3f.

Reviewer 3 Report

The present manuscript describes the findings of authors in the area of synthetic ways to substituted pyridazinodiquinoline-6,7-diones. The synthetic part is well performed. Also, the authors have provided the X-ray structure analysis of one compound (3a) and theoretical calculations rationalizing the formation of the syn - structure. Below I list questions and comments which should be addressed by authors.

  • The introduction looks interesting but more than 50% of references are based on the old literature (20-40 and up to 50 years old!). Please consider using newer references.
  • - Scheme 1 should be cited in the Introduction (page 2). And the number 6a in the title of this scheme is wrong.
  • The numeration of compounds on Scheme 2 and in the previous two paragraphs on page 3 (lines 73-90) is confusing, please check and correct accordingly. Eg. where are 5a–g in Scheme 2 as it is mentioned in line 80?; scheme 2 contains compounds of three different types but all have identical numbers 4a-g.
  • Also, scheme 2 contains compounds with subnumeration"a-g", but, at the same time, the reaction conditions are also numbered with "a-d". This creates a confuse. 
  • Next question to scheme 2: how can the required dimers 3a-g with N-R fragments be obtained from compounds 4a-g (in the centre of scheme 2), in which the nitrogen atom does not contain substituents R, by synthetic procedures c and e?
  • Scheme 3 creates a lot of questions. The main doubt is the formation of monooxygen radical O. in this reaction. Such radical is a very reactive strong oxidant species that should readily oxidize further any of the proposed intermediates. However, the authors say that the formation of electron-rich negatively charged intermediate 11a-g takes place in such conditions. All these stages should find an experimental confirmation. Without the latter, such a scheme is speculation only. 
  • The reference section: the last three references are wrongly combined into one reference (I mean refs. 54-56).
  • The section Conclusions should be seriously modified. In its current state, it is of little significance.

In general, as I understood from the content of the entire discussion of the results and the introduction, in fact, all the compounds described in this article are not new and have already been previously reported. And the novelty of the work comes down to varying the conditions for their synthesis and achieving higher preparative yields. From my point of view, in this regard, the manuscript does not meet the criteria of high-reputed journal Molecules. However, I would not like to reject this manuscript. Therefore I ask the authors to argue more strongly the novelty and importance of this work. They should make both the "Introduction" and "Results and Discussion" sections (as well as "Conclusions") more informative and meaningful.

Author Response

Answer: The introduction was modified with their references as 17 new updated refs were added (replacing the old refs). The coupling constants of substituted aromatic were written for compounds 3c, 3d and 3f. Moreover the DFT method of calculation was concerned on the stability of compound 3a, since formation of compound 3’a would be explained by different mechanism. That suggestion was also be asked by the other reviewers. The conclusion was also changed. Moreover the mechanism was changed according to the advice of other reviewer. We changed the mechanism totally to be more reasonable taking into consideration that proposed by literature [38].

All alternations were made by red highlights.

The present manuscript describes the findings of authors in the area of synthetic ways to substituted pyridazinodiquinoline-6,7-diones. The synthetic part is well performed. Also, the authors have provided the X-ray structure analysis of one compound (3a) and theoretical calculations rationalizing the formation of the syn - structure. Below I list questions and comments which should be addressed by authors.

  • The introduction looks interesting but more than 50% of references are based on the old literature (20-40 and up to 50 years old!). Please consider using newer references.
  • - Scheme 1 should be cited in the Introduction (page 2). And the number 6a in the title of this scheme is wrong.

Answer: New 17 updated new refs were added (the old ones were replaced). The numbering of compound 6a was also changed into 3a.

  • The numeration of compounds on Scheme 2 and in the previous two paragraphs on page 3 (lines 73-90) is confusing, please check and correct accordingly. Eg. where are 5a–g in Scheme 2 as it is mentioned in line 80?; scheme 2 contains compounds of three different types but all have identical numbers 4a-g.

Answer: All were corrected in red highlight (Also in the scheme itself).

  • Also, scheme 2 contains compounds with subnumeration"a-g", but, at the same time, the reaction conditions are also numbered with "a-d". This creates a confuse. 

Answer: Was corrected

  • Next question to scheme 2: how can the required dimers 3a-g with N-R fragments be obtained from compounds 4a-g (in the centre of scheme 2), in which the nitrogen atom does not contain substituents R, by synthetic procedures c and e?

Answer: That was also corrected in red highlights

  • Scheme 3 creates a lot of questions. The main doubt is the formation of monooxygen radical O. in this reaction. Such radical is a very reactive strong oxidant species that should readily oxidize further any of the proposed intermediates. However, the authors say that the formation of electron-rich negatively charged intermediate 11a-g takes place in such conditions. All these stages should find an experimental confirmation. Without the latter, such a scheme is speculation only. 

Answer: According to the advice of other reviewers, we change the mechanism totally to be more reasonable taking into consideration that proposed by literature [38].

  • The reference section: the last three references are wrongly combined into one reference (I mean refs. 54-56).

Answer: Was corrected and the total number of refs becomes 38, after addition of two recent refs related to pentacyclic heterocycles.

  • The section Conclusions should be seriously modified. In its current state, it is of little significance.

Answer: Was corrected

In general, as I understood from the content of the entire discussion of the results and the introduction, in fact, all the compounds described in this article are not new and have already been previously reported. And the novelty of the work comes down to varying the conditions for their synthesis and achieving higher preparative yields. From my point of view, in this regard, the manuscript does not meet the criteria of high-reputed journal Molecules. However, I would not like to reject this manuscript. Therefore I ask the authors to argue more strongly the novelty and importance of this work. They should make both the "Introduction" and "Results and Discussion" sections (as well as "Conclusions") more informative and meaningful.

Addition paragraphs were added in the introduction announced to the pentacyclic Nitrogen heterocycles. Moreover the conclusion was changed to the way to show the prospective interest of the obtained products, which might have important biological activities.

Round 2

Reviewer 1 Report

Necessary comments have been taken into account and corrected.

Author Response

Many thanks for your patience and best guidence

Reviewer 2 Report

The authors have made changes in accordance with the comments of the reviewer.

There are some comments:

1) It is necessary to remove terms "syn and anti isomers" because these terms refer to diastereoisomers.

2) The assignment of signals in the 13C NMR spectra should be removed because the authors' assignment is not based on 2D NMR experiments.

Author Response

There are some comments:

  • It is necessary to remove terms "syn and anti isomers" because these terms refer to diastereoisomers.

Answer: Was removed from abstract (red highlights)

  • The assignment of signals in the 13C NMR spectra should be removed because the authors' assignment is not based on 2D NMR experiments.

Answer: OK was removed (except the well-defined carbons like carbons of Methyl and carbonyl). The others were removed

Many thanks

Reviewer 3 Report

The authors have revised the manuscript in the accordance with all reviewers' comments. I believe it may be now accepted for further publication process. 

Author Response

Thanks

This manuscript is a resubmission of an earlier submission. The following is a list of the peer review reports and author responses from that submission.

Round 1

Reviewer 1 Report

Authors improved the paper entitled "Autoxidation of 4-hydrazinylquinolin-2(1H)-one; Synthesis of pyridazino[4,3-c:5,6-c']diquinoline-6,7(5H,8H)-diones" scientifically, and included my reservations  in an updated draft.

Reviewer 2 Report

The manuscript by S. M. Mostafa has been certainly revised, however I am afraid I cannot accept the manuscript for publication. There have been yet incorrect structures in Schemes. For example, while the reaction of 4a-g from 1a-g in Scheme 2 shows the chlorination accompanied by deoxygenation, why the 4a-g is reduced to dihydroquinoline using POCl3? Structure 3’a has not been corrected in Figure 3 (N-Et and C=O should be interchanged at right hand structure). I strongly recommend Scheme 3 must be deleted, otherwise be reconsidered advisedly because there are a lot of inaccuracies. For example, in the reaction from 7a-g to 8a-g, the arrow of 1,3-H shift is correct? I think that the zwitterion 8a-g should be converted to neutral form by elimination of N2 in intramolecular mechanism instead of intermolecular mechanism (4-non-substituted quinoline-one should be produced). Although there are many other points to correct, please have an expert check the mechanism carefully. Even though you mention the reaction optimization (Table 1), the detailed reaction condition should be shown (temperature and equivalent of base, etc.). While I pointed out in original manuscript, the discussion of structure 3’a is completely meaningless in this paper because the compound has not been experimentally obtained. Contrastingly, the dimerized structure 3a corroborated by X-ray data is interesting. As such, the manuscript must be polished and checked solidly before resubmitted to somewhere.

Reviewer 3 Report

Some comments have been taken into account and corrected.

However, the authors did not rewrite the conclusion in such a way that it reflects the main content of the article.

The references still need to be refreshed. The authors have been added only two new articles (2019 and 2020) while the most references date until 2015. Only 10 references from 55 date from 2015 and later.